# Prognostic Impact of PD-L1 Expression in pN1 NSCLC: A Retrospective Single-Center Analysis

**DOI:** 10.3390/cancers13092046

**Published:** 2021-04-23

**Authors:** Florian Eichhorn, Mark Kriegsmann, Laura V. Klotz, Katharina Kriegsmann, Thomas Muley, Christiane Zgorzelski, Petros Christopoulos, Hauke Winter, Martin E. Eichhorn

**Affiliations:** 1Department of Thoracic Surgery, Thoraxklinik, Heidelberg University, 69117 Heidelberg, Germany; laura.klotz@med.uni-heidelberg.de (L.V.K.); Hauke.Winter@med.uni-heidelberg.de (H.W.); martin.eichhorn@med.uni-heidelberg.de (M.E.E.); 2Translational Lung Research Center, German Center for Lung Disease (DZL), 69120 Heidelberg, Germany; Mark.Kriegsmann@med.uni-heidelberg.de (M.K.); Thomas.Muley@med.uni-heidelberg.de (T.M.); Petros.Christopolous@med.uni-heidelberg.de (P.C.); 3Institute of Pathology, Heidelberg University Hospital, 69120 Heidelberg, Germany; christiane.zgorzelski@med.uni-heidelberg.de; 4Department of Hematology, Oncology and Rheumatology, Heidelberg University, 69117 Heidelberg, Germany; Katharina.Kriegsmann@med.uni-heidelberg.de; 5Section Translational Research (STF), Thoraxklinik, Heidelberg University, 69117 Heidelberg, Germany; 6Department of Thoracic Oncology, Thoraxklinik, Heidelberg University Hospital, 69120 Heidelberg, Germany

**Keywords:** lung cancer, PD-L1, adjuvant therapy, surgery, immunotherapy, NSCLC

## Abstract

**Simple Summary:**

The analysis of prognostic biomarkers (e.g., PD-L1) helps to define treatment for lung cancer patients. To date, these markers have only been examined in metastatic or inoperable situations. We analyzed the PD-L1 expression-levels of tumors from 277 lung cancer patients that underwent curative intent surgery. PD-L1 was identified as a prognostic factor, depending on histologic subtype.

**Abstract:**

The programmed death-ligand 1 (PD-L1) plays a crucial role in immunomodulatory treatment concepts for end-stage non-small cell lung cancer (NSCLC). To date, its prognostic significance in patients with curative surgical treatment but regional nodal metastases, reflecting tumor spread beyond the primary site, is unclear. We evaluated the prognostic impact of PD-L1 expression in a surgical cohort of 277 consecutive patients with pN1 NSCLC on a tissue microarray. Patients with PD-L1 staining (clone SP263) on >1% of tumor cells were defined as PD-L1 positive. Tumor-specific survival (TSS) of the entire cohort was 64% at five years. Low tumor stage (*p* < 0.0001) and adjuvant therapy (*p* = 0.036) were identified as independent positive prognostic factors in multivariate analysis for TSS. PD-L1 negative patients had a significantly better survival following adjuvant chemotherapy than PD-L1 positive patients. The benefit of adjuvant therapy diminished in patients with PD-L1 expression in more than 10% of tumor cells. Stratification towards histologic subtype identified PD-L1 as a significant positive predictive factor for TSS after adjuvant therapy in patients with adenocarcinoma, but not squamous cell carcinoma. Routine PD-L1 assessment in curative intent treatment may help to identify patients with a better prognosis. Further research is needed to elucidate the predictive value of PD-L1 in an adjuvant setting.

## 1. Introduction

Lung cancer is the world’s leading cause of cancer-mortality [1]. Surgery plays a key-role in the treatment. More than one-third of patients experience a relapse of disease, even after radical tumor resection in combination with systematic mediastinal and hilar lymph node dissection [2]. Nodal metastases have been repetitively identified as a strong predictive factor for survival. Five-year survival ranges from 50% in N1-positive to 30% or less in patients with N2-disease [3,4]. To date, adjuvant chemotherapy is recommended as standard treatment for any pN-stage and results in a survival benefit of only approximately 5% [5,6]. The decision for adjuvant chemotherapy does not take the tumor histology or potential therapeutic targets (e.g., actionable mutation profiles) into account.

To date, therapies targeting the immune system are routinely used in patients with locally advanced or metastatic disease only [7,8,9]. Individualized systemic treatment concepts have not yet been transferred to early lung cancer treatment. However, observed relapse rates in up to two-thirds of the patients, even after complete tumor resection, underline the urge for novel multimodal therapeutic approaches in potentially curable tumor stages [10,11]. Recently, impressive tumor responses to neoadjuvant immunotherapy have been reported [12,13]. However, proof of the therapeutic efficacy of adjuvant immunotherapy is unknown with a lack of reliable results in large scale patient cohorts [14,15]. It is also unclear whether the expression of the programmed death-ligand (PD-L1), which is routinely used to predict the benefit from immunotherapy in non-operable non-small cell lung cancer (NSCLC) [16], is of prognostic relevance for adjuvant lung cancer treatment. Consequently, PD-L1 is not routinely analyzed in early lung cancer, nor does it currently guide the decision towards adjuvant treatment after surgery.

We reviewed surgical patients with a pN1 NSCLC. All tumors were immunohistochemically evaluated for PD-L1 expression on a tissue microarray (TMA). Data were analyzed for a possible prognostic impact of PD-L1 expression and other clinicopathological factors associated with outcomes in order to identify cases with a presumably higher susceptibility to immunomodulatory treatment. 

## 2. Materials and Methods

### 2.1. Patient Selection

Surgical specimens from 317 consecutive patients with pN1 NSCLC, who were treated at the Thoraxklinik in Heidelberg between January 2010 and December 2016, were reviewed by the author (FE) and the institutional pathologist (MK). All patients had undergone primary curative intent thoracic surgery (complete anatomical tumor resection in combination with systematic mediastinal and hilar lymph node dissection). Complete information on staging and follow-up were available from the institutional database. Patients with neoadjuvant chemotherapy or incomplete resections, as well as those who died independently from tumor progression within 90 days after surgery, were excluded from the analysis. The 7th edition of the TNM- system was used, as it was valid throughout the treatment period in determining primary surgical treatment and adjuvant therapy. 

The tissue was provided by the tissue bank of the National Center for Tumor Diseases (NCT, Heidelberg, Germany). The analysis was performed in accordance with the ethical regulations of the NCT tissue bank, defined by the local ethics committee (S-174/2019). Diagnoses were made according to the recommendations of the 2015 world health classification of tumors of the lung, thymus and heart [17]. Inclusion criteria were as follows: full availability of both tumor tissue and adjacent normal lung tissue and tumor cell quality for conclusive immunohistochemical staining and diagnostic processing (tumor cell content ≥10% tissue area). Finally, samples from 277 of 317 patients were eligible for further analysis. 

### 2.2. Tissue Processing and Immunohistochemistry

Tissue microarrays (TMA) were prepared as described previously [18,19]. In brief, two cylindrical cores measuring 1 mm in diameter from the tumor center, invasive margin and periphery were extracted from the donor block and subsequently transferred to a receiver block using an automatic TMA-roboter (TMA Grand Master, 3D Histech, Budapest, Hungary). Tissue sections were stained with the antibody clone SP263 (RTU, CE IVD) on an automated stainer (Benchmark XT, Roche, Mannheim, Germany). PD-L1 staining was evaluated according to current recommendations for lung cancer [20]. Results were recorded as the percentage of positive tumor cells for each TMA core (tumor proportion score). PD-L1 was determined positive if 1% of tumor cells were stained positive. Subsequent analysis evaluated cut-offs of ≥10% and ≥50% to assess the impact of PD-L1 at different proportion scores. The distribution of PD-L1 positive tumor cells was analyzed for both the tumor center and the invasive margin.

### 2.3. Statistical Analysis

Tumor-specific survival (TSS) was defined as the time from the date of surgery to the date of tumor-related death or last follow-up in censored, alive patients. Disease-free survival (DFS) was defined as the time from the date of surgery until the date of first detection of tumor relapse. Data were collected and analyzed using SPSS version 25 (IBM Corporation, Armonk, NY, USA). Survival was calculated using the Kaplan–Meier product method, while the log rank-test was used to calculate univariate differences. The Cox-regression model was used for multivariate analysis. Rates and proportions were analyzed using the chi-square or Fisher’s exact test. A *p*-value of less than 0.05 was considered statistically significant. 

## 3. Results

### 3.1. Clinical Patient Characteristics

Data from 277 patients (186 male (67.1%)), with a mean (±standard deviation (SD)) age of 65,1 (±9.7) years were retrospectively analyzed (Table 1). Lobectomy was performed in 178 patients (64.3%), bilobectomy in 18 patients (6.5%) and pneumonectomy in 81 patients (29.2%). There were 151 (54.5%) squamous cell carcinomas and 126 (45.5%) adenocarcinomas. 

174 patients (62.8%) received adjuvant platinum-based chemotherapy. Systemic treatment was not administered in 103 patients (37.2%). The main reasons for deciding against chemotherapy were comorbidities (68.9%) or refusal (25.2%).

### 3.2. PD-L1 Assessment

PD-L1 was found to be positive (≥1% positive tumor cells) in 146 patients (52.7%). Mean expression level was 20% (minimum 1%, maximum 100%). PD-L1 staining was found to be positive in 1–9% of tumor cells in 49 patients (17.7%), 10–49% of tumor cells in 44 patients (15.9%) and in 50–100% of tumor cells in 53 patients (19.1%). 

PD-L1 expression was more frequently observed in squamous cell carcinoma than adenocarcinoma (60.3% vs. 43.7%; *p* = 0.008) and more often in patients younger than 65 years (60.5% vs. 44.4%; *p* = 0.008). Positive PD-L1 status was more frequent in patients with better performance status; however, this was not significant (ECOG 0 vs. ECOG 1, 55.8% vs. 42.2%, *p* = 0.064). A higher rate of PD-L1 positive tumors was found in patients that received adjuvant chemotherapy compared to the subgroup without adjuvant therapy (58.0% vs. 43.7%; 0.025). PD-L1 was equally distributed throughout all tumor stages (52.6% (stage II) vs. 52.9% (stage III), *p* = 1.0). 

PD-L1 positive tumor cells were exclusively found in the tumor core in 19 of 146 patients, in the invasive zone in 8 of 146 patients and in both compartments in 119 of 146 patients (Figure 1). Among these, 25 patients (21.0%) showed a predominant distribution of PD-L1 positive tumor cells in favor of the tumor core and 64 patients (53.8%) of the invasive margin. Thirty PD-L1 positive patients (25.2%) showed an equal staining intensity. 

The calculated difference of PD-L1 expression levels in the tumor core and invasive zone was more than 1% in 51 patients (34.9%), more than 10% in 37 patients (25.3%), more than 25% in 13 patients (8.9%) and more than 50% in 15 patients (10.3%).

### 3.3. Prognostic Factors for Long-Term Survival and Relapse

The mean follow-up was 52 months with 169 (61.0%) of the patients still alive. TSS of the entire cohort was 64% at five years. Adjuvant chemotherapy significantly improved survival (70.0% vs. 54.2% at five years, *p* = 0.014) in the univariate analysis. Multivariate analysis of the entire cohort confirmed adjuvant chemotherapy (*p* = 0.036) and low tumor stage (*p* < 0.0001) as independent positive predictive factors for survival. A trend towards superior survival was observed for patients with squamous cell histology (*p* = 0.063) (Table 2).

Positive PD-L1 staining determined at a threshold of 1% showed a trend towards improved survival in the entire cohort (70.3% vs. 57.4%, *p* = 0.07) in univariate analysis (Figure 2).

Low tumor stage and adjuvant chemotherapy treatment were identified as individual factors with a significant positive impact on tumor-specific survival. Squamous cell histology showed a trend towards better outcomes. AC, adenocarcinoma; SCC, squamous cell carcinoma; CI, confidence interval; HR, Hazard Ratio.

The proportion of PD-L1 positive patients was higher in the chemotherapy cohort (58.0% vs. 43.6%, *p* = 0.025). 

Adjuvant therapy significantly improved survival in PD-L1 negative patients (*p* = 0.042). Patients with positive PD-L1 status (at a threshold of 1%) experienced only an insignificant trend towards improved survival by chemotherapy. Adjuvant chemotherapy did not improve survival of patients with positive PD-L1 immunostaining in more than 10% of the tumor cells (*p* = 0.995; Figure 3).

However, different prognostic trends were observed among distinct histologic subtypes. PD-L1 was identified as a positive predictive factor for chemotherapy response in patients with adenocarcinoma histology (*p* = 0.019), but not squamous cell carcinoma (*p* = 0.155, Figure 4a,b). The latter small subgroup showed a trend towards inferior survival by adjuvant chemotherapy in patients with a PD-L1 expression of >10%.

The multivariate analysis identified positive PD-L1 status (*p* = 0.007) and low tumor stage (*p* = 0.008) as independent prognostic factors for increased survival of patients with adenocarcinoma who received adjuvant chemotherapy (Table 3). Sex, younger age or better performance status were not associated with better survival in this subgroup.

Multivariate analysis in a subgroup of patients with adenocarcinoma histology and adjuvant treatment identified low tumor stage and positive PD-L1 expression as being independently associated with beneficial outcomes. CI, confidence interval. ECOG, Eastern Cooperative Oncology Group.

At the time of last analysis (January 2021), 140 patients (82.8%) were free of tumor recurrence. 114 patients (41.2%) of the entire cohort (277 patients) suffered from tumor relapse, and overall DFS was 57.7% at 5 years. Localized relapse was observed in 31 patients (27.2%), distant isolated metastases in 61 patients (53.5%) and multifocal or disseminated recurrence in 22 patients (19.3%). Adjuvant chemotherapy treatment did not influence the risk of recurrence (*p* = 0.325). High tumor stage (*p* < 0.001) and adenocarcinoma histology (*p* = 0.007) were identified as independent negative factors predicting recurrence in multivariate analysis. Differences in sex, PD-L1 status and age were not identified as predictive for disease relapse. After chemotherapy, adenocarcinoma histology was associated with a higher risk of recurrence than squamous cell carcinoma (*p* = 0.042). Positive PD-L1 expression was deemed beneficial in this patient subgroup; however, it was not statistically significant (*p* = 0.081).

## 4. Discussion

The impact of PD-L1 expression on tumor tissue from a well-characterized clinical cohort of 277 patients with N1-NSCLC on survival was retrospectively analyzed. 

Squamous cell histology, limited tumor stage and adjuvant chemotherapy were identified as positive predictive factors of survival for the entire cohort. However, subgroup analysis suggested a potential prognostic value of PD-L1 expression, specifically in patients with adenocarcinoma histology.

The extent of lymph node metastases strongly determines tumor staging and survival of lung cancer patients. In N1-NSCLC, primary surgery is the treatment of choice. However, indication for adjuvant standard chemotherapy insufficiently takes into consideration distinct prognostic subgroups that have been identified repetitively [3,21,22]. Different therapeutic options exist for the treatment of patients with metastasized NSCLC, depending on the histology and the biomarker-status. For instance, routine testing for PD-L1 is indispensable as checkpoint inhibitors have been approved for stage IV disease, either as mono- or combination first-line therapy [23,24]. A paradigm shift in early lung cancer treatment is of current interest, as the first promising results on adjuvant anti-PD1/-PD-L1 therapy upfront surgery have been reported [12,13]. However, the role of adjuvant anti-PD1 immunotherapy has yet not been adequately addressed, and few phase III trials have been registered so far, but final analyses are pending (NCT02504372, NCT02273375, NCT02595944, NCT02486718, clinicaltrials.gov, accessed on 13 March 2021).

How far PD-L1 qualifies as a prognostic biomarker is of great interest in the literature, and both positive [25,26,27,28] and negative associations [29,30,31] with survival have been reported. A recently published meta-analysis by Tuminello et al. included more than 10,000 surgical patients (stages I-III NSCLC), and high PD-L1 expression was associated with poor survival, statistically significant in adenocarcinoma, but not in squamous cell carcinoma [32]. The authors discuss a potential bias by lacking information on concurrent systemic therapies (adjuvant and neoadjuvant). Second, data on specific inclusion and exclusion criteria of the selected studies was missing in 50% of the patients.

Positive PD-L1-status showed a trend towards better outcomes in univariate analysis of our entire patient cohort. The proportion of positive PD-L1 patients was significantly higher in patients that were treated with adjuvant chemotherapy. Notably, adjuvant chemotherapy and low tumor stage were identified as significant beneficial factors in multivariate testing of all patients. However, positive PD-L1 was predictive for improved survival after chemotherapy in multivariate analysis of patients with adenocarcinoma histology, but not squamous cell carcinoma. A prognostic impact of PD-L1 in specific subgroups of surgical NSCLC has been also reported for patients with high marker expression (>50%) [26] or with N2-lmyph node metastases [28]. Schmidt et al. reported on improved overall survival in a subgroup with squamous cell carcinoma histology, lymph node metastases and adjuvant therapy [27]. Yang et al. analyzed 163 patients following resection of stage I adenocarcinoma and found a correlation between PD-L1 expression and prolonged relapse-free survival, but not overall survival [33].

The variety of results and different prognostic impact of PD-L1 in specific subgroups warrant a controversial discussion on PD-L1 as a predictive marker. It is important to realize that different anti-PD-L1 capture antibodies, different thresholds to determine PD-L1 positivity and individual characteristics, such as ethnicity or tumor histology, may affect prognostic estimations. For instance, different proportions of PD-L1-positive tumors for distinct histological NSCLC subtypes have been reported. A higher rate of PD-L1 positive cells in resected tumors with adenocarcinoma histology than squamous cell carcinoma was found by Mu et al. (65% vs. 44%, *p* = 0.032) [34]. However, others observed a higher proportion of PD-L1 positive tumors in surgical cohorts with squamous cell histology [27,29,35]. These results were in accordance with our series that identified a predominance of PD-L1 expression in squamous cell carcinomas. In this specific smoking-associated entity, the potential that accompanied smoking induced acquired somatic mutations along with oxidative stress that may result in the observed elevated expression of PD-L1 protein [36], thus leading to durable PD-1 antibody response in clinical studies [37,38,39].

The benefit of adjuvant chemotherapy in the entire N1 cohort diminished with increasing PD-L1 expression levels. Moreover, a trend towards inferior outcomes has been observed for patients with squamous cell carcinoma histology and elevated PD-L1. How far adjuvant immunotherapy can be alternatively administered has to be addressed in the future. However, the benefit of checkpoint-inhibitory therapy in PD-L1 positive (at any threshold) advanced and metastatic NSCLC is undisputed [24]. Our analysis identified elevated PD-L1 as a positive prognostic predictor in adenocarcinomas treated with adjuvant chemotherapy. In a neoadjuvant setting, chemotherapy has been found to induce changes in the immune microenvironment promoting antitumor immunity [40]. Consequently, further research is warranted to evolve potential synergistic effects of combination chemo-/immunotherapy and to identify subgroups susceptible for responses in multimodal treatment concepts with curative intent.

Differences in survival depending on patients’ ethnicity and race have been repetitively reported among various malignancies, including lung cancer [41]. For instance, the influence of heterogeneous prognostic characteristics, such as smoking status, prevalence of genomic alterations, and response to chemotherapy, were discussed on the course of the disease in a comparative study of Asian and non-Asian individuals [42,43]. Ethnic disparities were also apparent in two meta-analyses on the predictive role of PD-L1. Among these, PD-L1 expression was associated with poor survival. Of note, the majority of source data were from Asian patients [32,44]. Only a few series with Caucasian patients were included within adversely identifying PD-L1-expression as a favorable predictor [26,27,45]. The results of our series assimilate the existing European results; however, the complex pathomechanisms accountable for the observed ethnic disparities are under current investigation. Conclusively, a number of covariates, such as heterogeneity in the molecular landscape, must be deemed responsible for the observed differences [46,47].

The indication for immune checkpoint inhibitor-therapy in lung cancer is strongly linked with PD-L1 assessment. National and international definitions of PD-L1 testing in terms of used staining antibodies and thresholds for positivity quantitative analyses have been addressed and defined recently [20,48,49]. However, clinically approved positive cut-offs range from 1 to 50% and were inconsistently used in retrospective studies. The difficulty in determining a clinically appropriate threshold to define a positive, predictive test was discussed by Kerr et al. The authors propose avoiding very low staining thresholds (1–5%), as these would more likely inaccurately reflect the patient’s overall tumor burden because of intra-tumoral heterogeneity and a higher risk of inconsistencies in scoring [50]. These suggestions are supported by a series from Illie et al. which revealed major discordances between diagnostic biopsies and surgical specimens in 48% of cases. In all cases, the biopsy specimens underestimated the PD-L1 status observed on the whole tissue sample [51]. Heterogeneous PD-L1 expression was also found in our series, as positive staining was restricted to either the tumor core or the invasive margin in 18% of patients. Second, a deviation of 10% or more in staining extent was found in 44.5% of PD-L1 positive patients, depending on the analyzed region (tumor core or invasive margin). In a clinical scenario, diagnostic procedures (e.g., transbronchial biopsy, needle-core biopsy) should consider these findings and counter intra-tumoral heterogeneity by sequential biopsies from different tumor regions.

Standard adjuvant treatment for nodal positive patients comprises platinum-based combination chemotherapy irrespective of histological subtype or distinct predictive biomarkers. Chemotherapy failed to improve survival in our series in patients with a PD-L1 expression of 10% or more. However, this observation could be explained by opposed survival impact after stratification towards histological subtypes, with significant superior results in the adenocarcinoma subgroup.

The study has limitations: due to its retrospective design and the selected patient cohort, the results of PD-L1 analysis were not available at the timepoint of interdisciplinary decision towards adjuvant treatment. The observed trend towards better survival of PD-L1 positive patients in the entire cohort is assumed to be biased by the higher proportion of PD-L1 positive tumors in patients receiving chemotherapy, which was identified as a strong positive prognostic factor. However, analysis of all patients receiving standard adjuvant systemic treatment revealed PD-L1 protein analysis as potentially predictive for relevant subgroups of limited stage NSCLC.

## 5. Conclusions

PD-L1 expression was identified as a positive predictor for survival in patients with an adenocarcinoma histology receiving adjuvant chemotherapy. Moreover, a trend towards negative impact on survival was observed in PD-L1 positive patients with squamous cell histology. A prognostic relevance of PD-L1 expression in the adjuvant treatment course can be hypothesized. However, the sample size of our cohort was small and the clinical impact of our findings has to be further elucidated. Compared to the metastatic stage, systemic treatment following curative intent surgery should consider histology and PD-L1 expression status as potential prognostic factors towards individualized adjuvant therapy. Ongoing research in this context is urgently needed to address the utility of PD-L1 and other potential predictive factors in the postoperative setting.

## Figures and Tables

**Figure 1 cancers-13-02046-f001:**
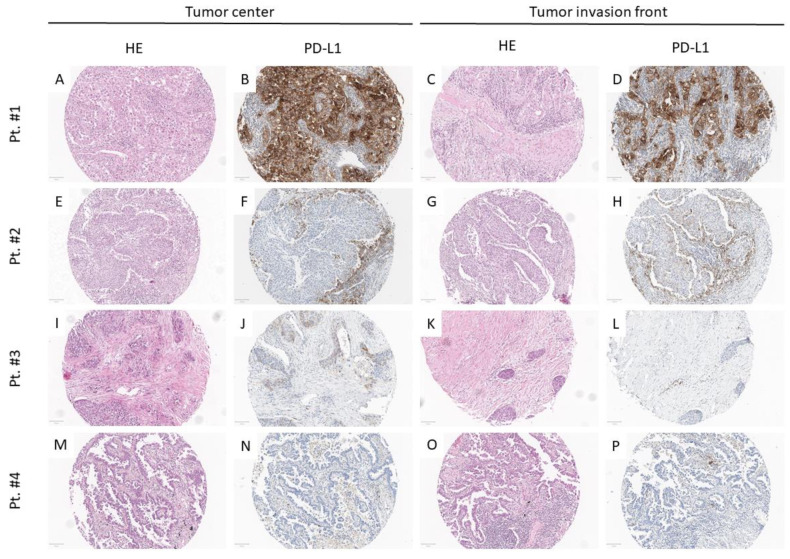
Examples of PD-L1 positive and PD-L1 negative tumors: Four patient samples (rows) from the tumor center (first and second column) and the respective invasion front (third and fourth column) stained with Hematoxylin and Eosin (**A**,**E**,**I**,**M**,**C**,**G**,**K**,**O**) and Programmed Cell Death Ligand 1—PD-L1; (**B**,**F**,**J**,**N**,**D**,**H**,**L**,**P**) are outlined. PD-L1 was either positive in the tumor center (**A**,**B**) and the invasion front (**C**,**D**), (pt. #1), at the invasion front (**G**,**H**) but not in the tumor center (**E**,**F**), (pt. #2), in the tumor center (**I**,**J**) but not at the tumor invasion front (**K**,**L**), (pt. #3) or negative in both (**M**–**P**), (pt. 4). Magnification: 100×, Scale bar: 100 µm.

**Figure 2 cancers-13-02046-f002:**
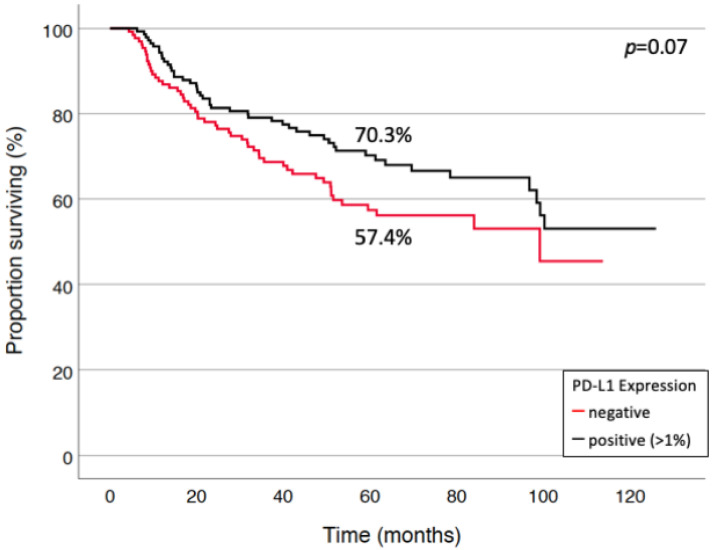
Tumor-specific survival by PD-L1 status. Patients with positive PD-L1 expression (*n* = 146) on tumor cells had a trend towards better 5-year survival (70.3% vs. 57.4%) in the entire cohort (*n* = 277).

**Figure 3 cancers-13-02046-f003:**
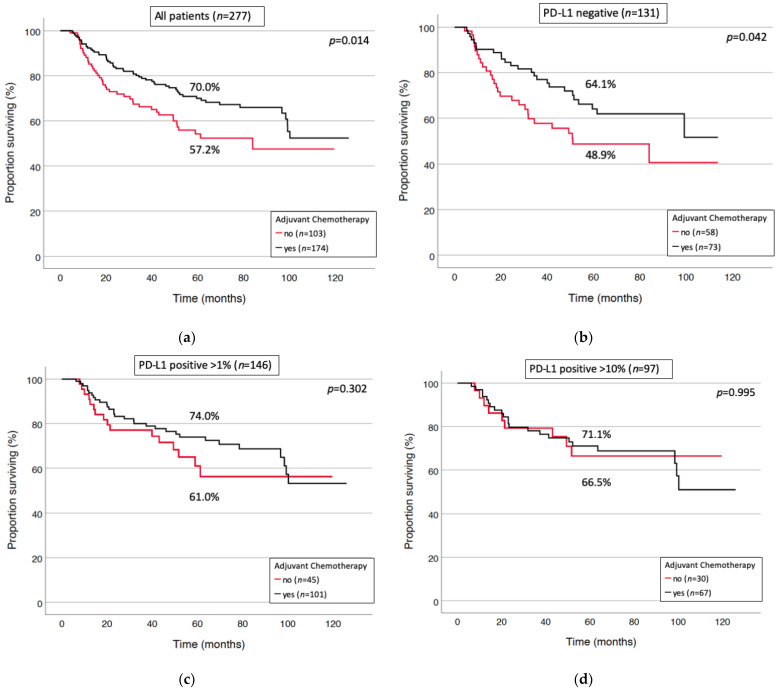
The effect of adjuvant chemotherapy in PD-L1 subgroups. Adjuvant chemotherapy improved survival in the entire study cohort (**a**) and in the PD-L1 negative subgroup (**b**). However, the beneficial effect of chemotherapy decreased in patients with PD-L1 expression >1% (**c**) and >10% (**d**).

**Figure 4 cancers-13-02046-f004:**
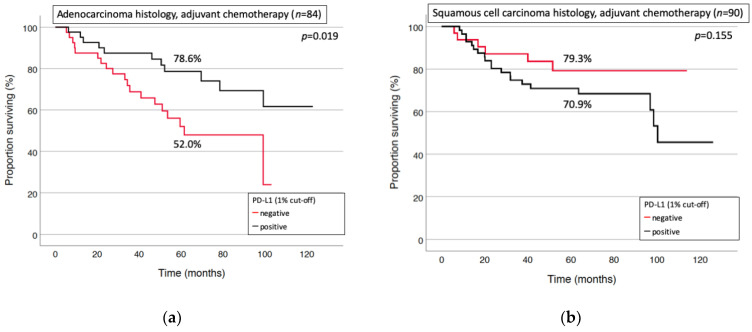
Positive PD-L1 status was associated with improved survival after chemotherapy in patients with adenocarcinoma (**a**), but not squamous cell carcinoma (**b**).

**Table 1 cancers-13-02046-t001:** Patient characteristics.

Variable	*n* (%)
No. of patients	277 (100%)
Gender	
Male	186 (67.1%)
Female	91 (32.9%)
Age, years: mean (range)	65.1 (40–89)
<65 years	142 (51.3%)
>65 years	135 (48.7%)
Performance status	
ECOG * 0	213 (76.9%)
ECOG 1	64 (23.1%)
Histology	
Squamous cell carcinoma	151 (54.5%)
Adenocarcinoma	126 (45.5%)
Surgical procedure	
Lobectomy	178 (64.3%)
Bilobectomy	18 (6.5%)
Pneumonectomy	81 (29.2%)
Tumor stage	
pT1a/b	25 (9.0%)
pT2a/b	131 (47.3%)
pT3	79 (28.5%)
pT4	42 (15.2%)
Adjuvant chemotherapy	
Yes	174 (62.8%)
No	103 (37.2%)
PD-L1 assessment	
Positive at any threshold	146 (52.7%)
>1% of Tumor cells positive	146 (52.7%)
>10% of tumor cells positive	97 (35.0%)
>50% of tumor cells positive	53 (19.1%)
negative	131 (47.3%)
First site of recurrence at Follow up	
No recurrence	163 (58.8%)
Local	31 of 114 (27.2%)
Single distant	61 of 114 (53.5%)
Multiple	22 of 114 (19.3%)
Tumor-specific survival (5-year; %)	64%
Disease free survival (5-year; %)	58%

* Eastern Cooperative Oncology Group.

**Table 2 cancers-13-02046-t002:** Multivariate analysis of prognostic factors for Tumor-Specific Survival.

Variable	Tumor-Specific Survival		
	HR	95% CI	*p*-Value
Histology SCC vs. AC	0.67	[0.43–1.02]	0.063
Adjuvant chemotherapy No vs. Yes	1.56	[1.03–2.37]	0.036
Tumor Stage IIA/IIB vs. IIIA	0.43	[0.28–0.64]	<0.0001
Sex Male vs. female	1.14	[0.74–1.78]	0.55
PD-L1—status Negative vs. positive (at any cut-off)	1.29	[0.86–1.93]	0.22
Age >65 years vs. <65 years	1.26	[0.84–1.94]	0.25

**Table 3 cancers-13-02046-t003:** Multivariate analysis of prognostic factors for TSS, subgroup adenocarcinoma and adjuvant chemotherapy.

Variable	Tumor-Specific Survival		
	HR	95% CI	*p*-Value
Tumor Stage IIA/IIB vs. IIIA	0.36	[0.17–0.77]	0.008
Sex Male vs. female	1.31	[0.61–2.80]	0.49
PD-L1—status Negative vs. positive (at any cut-off)	2.92	[1.33–6.39]	0.007
Age >65 years vs. <65 years	0.75	[0.34–1.66]	0.48
Performance status ECOG 0 vs. ECOG 1	1.66	[0.38–7.29]	0.50

## Data Availability

Data are available on request due to privacy and ethical restrictions.

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
