# Peer review of "Prognostic Impact of PD-L1 Expression in pN1 NSCLC: A Retrospective Single-Center Analysis"

_cancers, 2021, doi:10.3390/cancers13092046_

Round 1
Reviewer 1 Report
Praise to the authors. The manuscript is written very clearly, well structured. The factual information and discussion is excellent. Important and interesting findings. They are significant pending the results of clinical trials of adjuvant immunotherapy.
Minor concerns:
- The manuscript does not provide information about EGFR, ALK, etc. status. Because all patients were consistently included, it is unlikely that there were no positive patients for these markers.
- It is not formally stated that the subjects were not treated with target therapy, angiogenesis inhibitors. I guess immunotherapy was not given also.
Author Response
Reviewer 1
Praise to the authors. The manuscript is written very clearly, well structured. The factual information and discussion is excellent. Important and interesting findings. They are significant pending the results of clinical trials of adjuvant immunotherapy.
Dear reviewer,
We appreciate your response and thank you for your valuable comments. Please find our response below.
Minor concerns:
Point 1: The manuscript does not provide information about EGFR, ALK, etc. status. Because all patients were consistently included, it is unlikely that there were no positive patients for these markers.
Response to Point 1: A total of 277 patients was found eligible for final analysis. Only patients with pN1 were included. There were no patients that underwent neoadjuvant treatment. According to current guidelines, surgical tumor resection is the primary treatment in this patient cohort (update in section material and methods, line 74-76) This also accounts for EGFR or ALK positive tumors. To date, there is no neoadjuvant or adjuvant EGFR- or ALK-TKI therapy authorized in the examined patient cohort, therefore EGFR/ALK status was not available.
Point 2: It is not formally stated that the subjects were not treated with target therapy, angiogenesis inhibitors. I guess immunotherapy was not given also.
Response to Point 2: Primary treatment for all the patients was surgery plus adjuvant platinum-based chemotherapy if indicated after oncological review. A total of 103 patients did not undergo adjuvant chemotherapy, mainly due to relevant comorbidities or refusal. Other therapies were not administered in the first treatment course. We added an explanatory sentence to the manuscript (Results section, line 122).
In case of recurrence of disease, patients underwent complete restaging procedures and redo-biopsy. Subsequent therapy was then administered depending on the results. In case of relapse, all types of therapies and modalities, ranging from redo-surgery or local radiotherapy (in limited stage) or multi-drug treatment (chemotherapy, immunotherapy, targeted therapy) were allowed.

Reviewer 2 Report
Dear authors,
thank you very much for this sophisticated research. I have some comments and suggestions.
Introduction:
Please be more clear about the treatment indications of surgery: In which tumour stages do you use surgery, chemotherapy (and radiotherapy)?
56: Consequently, PD-L1 it is not routinely analyzed in early lung cancer nor does it currently guide the decision towards adjuvant treatment after surgery.
Methods:
Was there adjuvant RT applied in any of the patients?
Results:
What was the distribution of PD-L1 pos./neg. patients in relation to other prognostic markers (low tumour stage, younger patients, ECOG score?) and did you do any matching? PD-L1 was more often positive in younger patients, isn't that a bias?
128: more often in patients younger than 65 years: please provide percentages
Discussion:
Please be clearer about treatment regimes especially radiotherapy and chemoradiotherapy. In N1 tumors, surgery is not always a must, IIIA (T4 N1) requires CRT + Durvalumab or CRT + surgery or induction chemo + surgery + RT; IIIA1 and IIIA2 require a RT of the mediastinum after surgery + chemo.
You provide a lot of information about the current literature on PD-L1 expression. This is important, but should be shortened. I suggest a more profound discussion of your own findings: what could be a reason for the better prognosis of PD-L1 pos. patients in different subgroups / histological type? Why might checkpoint inhibition and chemotherapy be synergistic? What could be reasons for the inferiority of adding adjuvant chemo in PD-L1 positive SCCs?
248: suggests a favorable outcome of in case of high PD-L1 expression
277: alterations, response to chemotherapy) were discussed
Summary and advices:
The whole study reads as if RT does not exist at all in the treatment of NSCLC, although it is a very important treatment modality. Please add this in the introduction and discussion. Please add in the results section if patients in your cohort were irradiated.
Also, it would be important to know the ECOG score of the patients and to have a clear statement about the distribution of patients with positive and negative PDL1 score to see if the patient characteristics are equally distributed. Perhaps you could add a table that addresses this.
Your results are very interesting and provide a basis for more in-depth research on this topic. That said, you don't really address your findings in the discussion section, so perhaps you could add some thoughts of your own on your findings here.
Author Response
Reviewer 2
Dear authors,
thank you very much for this sophisticated research. I have some comments and suggestions.
Dear reviewer,
We appreciate your response and thank you for your valuable comments. Please find our response below.
Introduction:
Please be more clear about the treatment indications of surgery: In which tumour stages do you use surgery, chemotherapy (and radiotherapy)?
Primary surgery is indicated according to the actual treatment guidelines that were valid throughout the observational period. In brief, patients with clinical stage NSCLC ranging from N0 to N2 (resectable, single station involved, stage IIIA3) underwent primary surgery.
Our current analysis focused only on pN1 NSCLC and curative intend surgery was the primary treatment. Exclusion criteria were neoadjuvant treatment, incomplete resections and tumor independent deaths within the first 90 postoperative days. The material and methods section was updated (lines 74-76)
Adjuvant chemotherapy was debated in all patients, however not given for several reasons (mainly refusal, relevant comorbidity or delayed postoperative recovery (lines 123-125)).
Adjuvant radiotherapy was not administered as the series did not contain patients with residual microscopic tumor following resection nor mediastinal nodal N2- lymph node metastases.
56: Consequently, PD-L1 it is not routinely analyzed in early lung cancer nor does it currently guide the decision towards adjuvant treatment after surgery.
Thank you for your attention, we corrected the typo error.
Methods:
Was there adjuvant RT applied in any of the patients?
As incomplete tumor resections were initially excluded from the analysis, local radiotherapy was not administered as adjuvant treatment.
Results:
What was the distribution of PD-L1 pos./neg. patients in relation to other prognostic markers (low tumour stage, younger patients, ECOG score?) and did you do any matching? PD-L1 was more often positive in younger patients, isn't that a bias?
We retrospectively analyzed patients with pN1 NSCLC that underwent surgery in our institution. PD-L1 status was evaluated for its potential prognostic impact. A matching was not performed assuming that relevant patient characteristics, treatment decision and were equally distributed throughout the treatment period. Following your interesting comment, we reanalyzed data providing more details on correlation of PD-L1 to other prognostic markers. (Results section, lines 138-142).
128: more often in patients younger than 65 years: please provide percentages
Additional information was added to the text as well as to the patients characteristics table 1, line 137.
Discussion:
Please be clearer about treatment regimes especially radiotherapy and chemoradiotherapy. In N1 tumors, surgery is not always a must, IIIA (T4 N1) requires CRT + Durvalumab or CRT + surgery or induction chemo + surgery + RT; IIIA1 and IIIA2 require a RT of the mediastinum after surgery + chemo.
Our reported series focuses on surgically treated patients with pN1 NSCLC. Patients were treated from 2010 to 2016 according to the then- in current guidelines (PMID: 20217630) with TNM7 for stage determination. In this period, surgery was indicated in stage T1-3N1 and selected T4N1 in case of functional and technical operability (https://doi.org/10.1093/annonc/mdv187). However, T4-tumors with invasion of the spine or neurovascular apical bundle (superior sulcus tumors, Pancoast) or preoperative proven N2-disease (stages IIIA1 and IIIA2) were not included in our series.
The today valid NSCLC treatment recommendations expand therapeutic options to more individual strategies in stage III (including definite CT/RT and or IT Durva) depending on specific situations (compare www.onkopedia.com, section NSCLC // german S3-Guideline for Lung cancer, http://leitlinienprogramm-onkologie.de/Lungenkarzinom.98.0.html (las access: 14.04.2021)
In brief, we believe that stage T4N1 NSCLC (non-superior sulcus) should be treated by primary surgery if technically and functionally feasible, following adjuvant therapy according to current guidelines.
You provide a lot of information about the current literature on PD-L1 expression. This is important, but should be shortened. I suggest a more profound discussion of your own findings: what could be a reason for the better prognosis of PD-L1 pos. patients in different subgroups / histological type? Why might checkpoint inhibition and chemotherapy be synergistic? What could be reasons for the inferiority of adding adjuvant chemo in PD-L1 positive SCCs?
Thank you for your valuable comments, we shortened the literature and restructured the discussion section (lines 323-334).
The prognostic implication of elevated PD-L1 in specific subgroups has been reviewed. A higher proportion of PD-L1 positive patients with squamous cell histology have been reported. Differences in PD-L1 expression among histologic subgroups have been discussed in light of the underlying reasons taking into account variations in somatic mutations and distribution among different ethnicities.
The potential synergistic effect of chemo- and immunotherapy has been discussed taking into account therapy-related changes to the immune-microenvironment resulting in a cellular milieu more susceptible to checkpoint-therapy.
We moreover discussed the potential limitation of our series in light of potential confounding variables with influence on patients postoperative outcome.
248: suggests a favorable outcome of in case of high PD-L1 expression
277: alterations, response to chemotherapy) were discussed
Summary and advices:
The whole study reads as if RT does not exist at all in the treatment of NSCLC, although it is a very important treatment modality. Please add this in the introduction and discussion. Please add in the results section if patients in your cohort were irradiated.
Also, it would be important to know the ECOG score of the patients and to have a clear statement about the distribution of patients with positive and negative PDL1 score to see if the patient characteristics are equally distributed. Perhaps you could add a table that addresses this.
Your results are very interesting and provide a basis for more in-depth research on this topic. That said, you don't really address your findings in the discussion section, so perhaps you could add some thoughts of your own on your findings here.
